# Construction of a Tandem Repeat Peptide Sequence with Pepsin Cutting Sites to Produce Recombinant α-Melanocyte-Stimulating Hormone

**DOI:** 10.3390/molecules26206207

**Published:** 2021-10-14

**Authors:** Dai-Lin Jiang, Chao-Ling Yao, Nien-Jen Hu, Yung-Chuan Liu

**Affiliations:** 1Department of Chemical Engineering, National Chung Hsing University, Taichung 402, Taiwan; w321365@gmail.com; 2Department of Chemical Engineering, National Cheng Kung University, Tainan 701, Taiwan; yao8@gs.ncku.edu.tw; 3Graduate Institute of Biochemistry, National Chung Hsing University, Taichung 402, Taiwan

**Keywords:** α-melanocyte-stimulating hormone, tandem repeat peptides, antimicrobial peptide, pepsin cleavage, recombinant expression

## Abstract

The production of α-melanocyte-stimulating hormone (α-MSH), a peptide hormone composed of 13 amino acids, is attempted by recombinant expression using *E. coli* as the host. To achieve this aim, a synthetic gene containing eight tandem repeats of *msh* gene (8*msh*) was designed for ribosomal synthesis of 8 α-MSH. The merit of the strategy is to diminish the peptide toxicity against the host cell and to achieve a higher production yield. Pepsin cleavage sites are introduced between the peptides for enzymatic proteolysis to obtain the monomeric peptide of α-MSH. The constructed plasmid was transformed into different strains of *E. coli* hosts, and *E. coli* XL1-Blue with gene 8*msh* revealed the highest yield of 8 α-MSH. Although 8 α-MSH was fractionalized in the insoluble pellets after cell lysis, pepsin cleavage was able to produce soluble α-MSH peptide, as analyzed and confirmed by mass spectrometry and peptide activity assays. The production of α-MSH was quantified using HPLC with a yield of 42.9 mg/L of LB culture. This study demonstrates the feasibility of producing α-MSH using recombinant expression of tandem repeat gene. The production procedure involves minimal post-treatment and processing and can be scaled up for industrial application.

## 1. Introduction

Antimicrobial peptides (AMPs) are ribosomally synthesized peptide fragments widely distributed in mammals, amphibians, plants, and insects, constituting a natural defense system against microbes [1,2,3]. Cecropins are one of the most well-known antimicrobial peptides originally isolated from the chrysalis of the celestial silkworm [4,5]. Up to date, numerous AMPs have been identified, and novel applications of these peptides have been proposed and implemented. For example, AMPs can be utilized to replace the conventional chemical preservatives in the food processing industry [6,7]. AMPs also provide an alternative strategy to tackle the emergence of multidrug-resistant pathogens and reveal an enormous potential in clinical pharmacology and agriculture [8,9].

Isolating AMPs from a natural source is cost-ineffective due to poor abundance and laborious procedure of purification [10,11]. Nevertheless, the production of AMPs using recombinant expression remains challenging because of the short length and cytotoxicity. Consequently, obtaining the short peptides for research purpose exclusively depends on chemical synthesis [12,13,14]. In any case, the above-mentioned approaches have respective limitations for large-scale production of AMPs for industrial practice [15].

Melanocyte-stimulating hormones (MSHs) are a family of peptide hormones [16] derived from pro-opiomelanocortin (POMC), a polypeptide precursor with 241 amino acids produced at the pituitary gland. POMC is further cleaved to produce various peptide hormones, including adrenocorticotropic hormone (ACTH), β-endorphin, and various types of MSHs: α-MSH, β-MSH, and γ-MSH [17]. It has been demonstrated that α-, β-, and γ-MSHs have specific effects on the brain, and therefore, MSHs are also termed “neuropeptides” [18].

α-MSH consists of 13 amino acids produced by the arcuate nucleus neurons. Its main function is to induce melanin production, and it also exerts antipyretic, anti-inflammatory [19], neuroprotective, and immunomodulatory effects by interacting with various melanocortin receptors. Although α-MSH can stimulate the formation of melanin in response to UV, it plays other physiological roles, the most important of which include maintaining the overall balance of the skin [20], protecting cells from oxidative damage, and stimulating anti-inflammatory response [21]. It is also suggested that α-MSH has antimicrobial activity [22,23,24], acting as a defense barrier in the skin immune system [25].

Chemical synthesis of α-MSH has been utilized for research-based purposes [26,27]. However, mass production of the peptide using this strategy remains challenging due to the yield and cost. Here, we aim to develop a protocol to produce a recombinant polypeptide containing eight tandem repeats of α-MSH (encoded from the synthetic gene, 8*msh*, consisting of eight repeats of msh open reading frame), where the pepsin cleavage site is introduced between each α-MSH repeat. The expression condition for 8 α-MSH was optimized, and α-MSH was obtained by pepsin digestion. The identity of cleaved peptide fragments was validated by mass spectroscopy. The antimicrobial activity of α-MSH peptide was examined using *E. coli* inhibition assay. Furthermore, the melanin stimulating activity of α-MSH was evaluated by measuring the melanin content of melanoma cells. The study demonstrates a successful example of producing the short peptide of α-MSH using recombinant DNA technology and protein engineering. The production procedure can readily be scaled up with significantly improved cost-effectiveness.

## 2. Results and Discussion

### 2.1. Construction, Expression, and Extraction of 8 α-MSH

The DNA fragment 8*msh* encoding eight tandem repeats of α-MSH with pepsin recognition sites were codon-optimized and synthesized to achieve better translation efficiency in *E. coli* host cells. Pepsin reveals a broad sequence specificity for cleavage, including Phe-Val, Gln-His, Glu-Ala, Ala-Leu, Leu-Tyr, Tyr-Leu, Gly-Phe, Phe-Phe, and Phe-Tyr [28,29]. In this study, we utilized Gly-Phe as the cleavage site because the sequence motif is not present in the sequence of α-MSH. The 8*msh* gene sequence was subcloned into the pQE60 vector to construct the expression plasmid pQE60-8*msh*, as shown in Figure 1.

To test the hosts’ effect, different *E. coli* strains, including DH5α, JM103(DE3), BW25113, EPI300, W3110, HB101, XL1-Blue, ER2566, and BL21(DE3), were selected for expression screening. After plasmid transformation, the transformants were subjected to a colony PCR amplification using the primers in Table 1, and the results are summarized in Table 2. All the hosts show correct PCR products except for DH5α. For further confirmation, we extracted the plasmids from the hosts and performed restriction enzymes cleavage using BamHI and BglII, and we only succeeded in obtaining the DNA fragments with the correct size in XL1-Blue, ER2566, and BL21(DE3) *E. coli* cells. This is probably due to the toxicity of 8 α-MSH or DNA replication error induced by the highly repetitive sequence.

The molecular weight of the gene fragment 8*msh* encoding 8 α-MSH with C-terminal 6xHis-tag is 402 bp, and the expected molecular weight of the translated product is 16,139 Da. For overexpression trials, we added 0.9 mM IPTG at OD_600_ of 0.7 and incubated at either 15 °C or 37 °C for 16 h. A protein band with an approximate molecular weight of 16 kDa obviously appears, and the host X1-Blue at 37 °C incubation reveals the best protein yield (Figure 2). In addition, the majority of the target protein is in the fraction of pellet after centrifuging the lysate, suggesting overexpressed target protein forms insoluble inclusion bodies.

We then attempted to dissolve the insoluble fraction using various concentrations of urea (100 μL, 2–6 M urea in Tris-HCl buffer pH7.0). After urea treatment, only a very low level of 8 α-MSH fusion protein could be dissolved from the pellets even at 6M urea (Figure 3). The low dissolution yield hindered us from the subsequent dialysis and purification. However, it was noticed that the original inclusion body in the cell lysate pellets revealed a satisfying purity (Figure 3, Lane 3). Thus, we speculated that pepsin might cleave the insoluble 8 α-MSH in the inclusion body and release α-MSH peptide with better solubility.

### 2.2. Identification of 8 α-MSH Fusion Protein

To validate the protein identity of 8 α-MSH as observed in the pellet fraction on SDS-PAGE gel (Figure 3), in-gel digestion of the target protein using trypsin was performed, followed by tandem mass spectrometer system LC/MS/MS (TripleTOF^®^ 6600 System, Applied Biosystems Sciex). Referring to the analysis of the experimental masses of peptide fragments, four species of peptides corresponding to the expected constituents of the trypsin-digested 8 α-MSH tandem repeat can be identified (Figure 4). The evidence indicates that the tandem repeat protein 8 α-MSH can be correctly produced in the above-mentioned cultivation conditions in spite of the extremely poor protein solubility.

### 2.3. Pepsin Cleavage Reaction

Although a high concentration of urea failed to extract 8 α-MSH in an aqueous solution, we speculated that pepsin digestion might result in the cleavage of 8 α-MSH and produce short α-MSH peptides, which may ameliorate the solubility. To test this hypothesis, we harvested the cells of *E. coli* XL1-Blue overexpressing 8*msh*, followed by sonication cell lysis and centrifugation to obtain the lysate pellets. The pellets were subjected to pepsin treatment at pH 2, 37 °C for different incubation times. The enzymatic digestion was terminated by heat treatment at 100 °C for 10 min. Due to the antimicrobial activity of α-MSH, the antimicrobial activity of the cleaved product was monitored using *E. coli* inhibition assay. As a control, the cell lysates containing full-length 8 α-MSH was applied to the inhibition agar test, and no inhibition activity was revealed, providing a likely reason why the expression of 8*msh* is nontoxic to the host cells. In addition, the antimicrobial activity of pepsin cleavage buffer was also assayed, indicating nearly no background activity. For the lysate samples with pepsin treatment, it is noticed that an inhibition zone appears between 0 to 90 min of pepsin cleavage, and the highest inhibition zone is observed at 60 min (Table 3). Longer incubation of cleavage (90 min) shows a decreased inhibition activity, and no inhibition effect is shown at 120 min. This is likely because peptides were intensely digested to smaller peptides, resulting in a weaker antimicrobial activity.

To further verify the antimicrobial activity is produced by pepsin digestion, we tested the inhibition activity of pepsin reactions at different pH (Table 4). As it is well known that pepsin shows the best peptide cleavage activity at acidic pH, the antimicrobial inhibition activity of pepsin treated 8 α-MSH reveals the highest antimicrobial activity at pH 2. The results indicate that although 8 α-MSH fusion protein is mainly produced in the insoluble pellet, pepsin treatment can digest 8 α-MSH at the cleavage sites and release short α-MSH peptides in aqueous solution with the antimicrobial activity as expected.

### 2.4. Identification of α-MSH after Pepsin Cleavage

The cleaved 8 α-MSH (at 60 min) was subjected to peptide identification using a matrix-assisted laser desorption time-of-flight mass spectrometer system (MALDI-ToF MS, Voyager DE PRO, Applied Biosystems). The expected mass of α-MSH peptide along with the residual residues at the pepsin cleavage site is 1826.86 Da. In the MALDI-ToF analysis, a dominant peak representing a nominal mass (Mr) of 1827.87 Da can be identified (Figure 5a). The mass spectrometer data provide solid evidence showing the existence of α-MSH peptide, which accounts for the antimicrobial activity. The yield of α-MSH peptides after the pepsin reaction was further subjected to an HPLC assay with the synthesized peptide as the standard (Figure 5b), where a concentration of 40 mg/L was obtained using the culture conditions as detailed in Section 3.2.

### 2.5. Antimicrobial Activity of α-MSH

The antimicrobial activity of 8 α-MSH tandem repeat protein with different proteolysis incubation times was further analyzed to determine the lethal dose as described in Section 3.5, with the chemically synthesized α-MSH as the control. The median lethal dose (LD_50_) was determined on the basis of Finney’s method [30] by counting the number of colony (survival rate), and the results are shown in Figure 6 and Table 5. Among the samples with different incubation times of pepsin, 60 min pepsin digestion of 8 α-MSH reveals the most effective antimicrobial activity (LD50 = 7.19 × 10^−3^ mg/L), closely similar to the standard α-MSH peptide (LD50 = 1.28 × 10^−2^ mg/L). Pepsin digestion with shorter (0 and 30 min) or longer (90 and 120 min) times show inferior antimicrobial activity, in great agreement with the inhibition assay (Table 4). It is noticed that the concentration of pepsin-digested 8 α-MSH was adjusted to 40 mg/L, same as the standard α-MSH peptide, suggesting the α-MSH peptide from pepsin reaction shows a comparable antimicrobial activity as a standard peptide.

### 2.6. Melanin Stimulation Assay of α-MSH

To evaluate that the toxicity of cleaved peptides on mammalian cells, a cytotoxicity test using mouse melanoma cells B16-F10 was performed. The test samples are divided into the following two categories: a synthesized α-MSH standard and the α-MSH peptide obtained from pepsin cleavage at pH 2.0, 60 min. The cytotoxicity results are shown in Figure 7a, where the addition of blank buffer only was used as the benchmark. None of the growth inhibitory effects of standard and α-MSH samples exhibited on the cells were found. As expected, the overall tests confirm that α-MSH samples are not toxic to mouse melanoma cells B16-F10 [31,32].

Different concentrations of α-MSH (0, 10, 50, 100, 200, and 500 ng/mL) were used to stimulate mouse melanoma B16-F10 cells to produce melanin. The melanin content treated with buffer (α-MSH = 0) is used as the background, where the percentage of melanin content in the cells is set as 100%. The percentage of melanin produced by cells treated with α-MSH samples and standard was determined by the relative melanin content ratio measuring the absorption at a wavelength of 405 nm. As shown in Figure 7b, α-MSH can stimulate melanin production in a concentration-dependent manner, and pepsin cleaved α-MSH peptides show a lower but comparable stimulative effect as the chemically synthesized α-MSH. This weaker effect of pepsin cleaved α-MSH peptides might be because the purity of the cleaved peptides is not as high as the standard. The purity calculated on the basis of bioactivity in comparison with the standard is approximately 90% (89.5% ± 0.3%). These cell-based data demonstrate that the α-MSH using recombinant expression and pepsin digestion has similar melanin stimulating activity as that of the synthesized α-MSH standard.

### 2.7. Comparison of Antimicrobial Peptide Production

With the success of producing α-MSH peptide revealing similar activity as the standard, we intend to survey the previous literature and summarize the production, post-treatment, and yield of other recombinant antimicrobial peptides (Table 6). As the peptides are toxic to the hosts, protein fusion with the peptides is a practical solution to decrease cytotoxicity. It is noted that most of the recombinant fusion proteins have to go through various separation and purification processes to obtain the target peptide. His-tag fusion at the protein termini followed by nickel ion-immobilized metal ion affinity chromatography (Ni-IMAC) is the most commonly used purification strategy. However, it is a labor- and time-consuming procedure and may not be suitable for large scale production at the industrial level.

In this study, overexpression of 8 α-MSH tandem repeat gene in *E. coli* surprisingly produced insoluble fusion protein. However, the insoluble inclusion body reduced the cytotoxicity of *E. coli* host cells. Furthermore, beyond our expectation, the inclusion body revealed rather high purity and a direct pepsin treatment successfully cleaved tandem repeat peptides and released α-MSH into the aqueous phase. The unprecedented procedure circumvents the step of protein purification, enhancing the cost-effectiveness for industrial practice. On top of that, the tandem octameric expression of α-MSH masks the cytotoxicity for *E. coli* hosts and significantly increases the stoichiometry of the α-MSH peptide via pepsin digestion. The study presents a successful case of producing a functional peptide by recombinant fusion protein expression and may provide an alternative strategy for peptide production. The recombinant peptides show lower but comparable physiological activities (approximately 90%) with the chemically synthesized counterpart, and most importantly, the peptide production strategy can be straightforwardly scaled up to meet industry requirements.

## 3. Materials and Methods

### 3.1. Bacterial Strains, Synthetic 8 α-MSH Plasmids, and Chemicals

The tandem repeat gene, 8*msh*, was codon-optimized, synthesized by Shenggong Co., Ltd. (Taipei, Taiwan), and subcloned into a pQE60 expression vector. The synthesized peptide α-MSH was purchased from Genomics BioSci & Tech. Co., Ltd (Taipei, Taiwan). Pepsin from porcine gastric mucosa was purchased from Sigma-Aldrich (St. Louis, MO, USA) (≥250 units/mg). *E. coli* strains including DH5, JM109(DE3), BL21(DE3), XL1-Blue, ER2566, BW25113, EPI300, W3110, and HB101 were used as the expression hosts in this study. The genotype and relevant features are listed in Table 7. The primers used in this study are listed in Table 1. Other reagents are at analytical grade and obtained from local suppliers.

### 3.2. E. coli Cultivation

The plasmid transformed *E. coli* strains were cultivated as follows. The seed culture was prepared with inoculation of LB medium containing 50 μg/mL ampicillin (AM) with 3 loops of single colonies and cultivated overnight at 37 °C at the rotation speed of 200 rpm. The seed culture was used to inoculate the main culture with the ratio of 1% (*v*/*v*), and the culture was incubated at 37 °C, 200 rpm to reach a cell density (optical density OD 600 nm) of 0.7. Gene expression was induced by the addition of isopropyl β-d-1-thiogalactopyranoside (IPTG) to a final concentration of 0.9 mM. The cultures were further incubated at 37 or 15 °C, 200 rpm for 16 h. Then, 50 mL of the overnight culture was subjected to centrifugation (7500× *g*, 4 °C, 10 min) to obtain the supernatant and cell pellet. The pellet was resuspended in Tris-HCl buffer (20 mM Tris-HCl, pH 7.0, 250 mM NaCl) to the final optical density (OD_600_) of 10. Then, it was subjected to sonication for cell lysis, followed by centrifugation (13,000× *g*, 4 °C, 1 min) to fractionalize to the cell lysate into the supernatant and pellet.

### 3.3. Pepsin Cleavage

The pellets of cell lysate obtained from the harvested culture were dissolved in 25 mL of the cleavage buffer (0.4% (*w*/*v*) pepsin, 20 mM Tris-HCl pH 2.0 and 250 mM NaCl). Pepsin cleavage reaction was performed at 37 °C, 200 rpm in a shaker incubator. The samples were taken every 30 min and subjected to heat treatment to terminate proteolysis at 100 °C for 10 min. Due to the antimicrobial activity of α-MSH, the pepsin cleaved samples were subjected to the antimicrobial assay as described in Section 3.5 to monitor the cleavage efficiency [39,40].

### 3.4. Assays

Protein analysis was carried out using 15% SDS-PAGE and Coomassie Blue staining [41]. The expression protein level was quantified via the densitometry analysis of each band on SDS-PAGE gels using ImageJ (version 1.49 s) [42]. Pepsin cleaved α-MSH was analyzed using an HPLC system using the SCpak C8-BIO column (250 mm × 4.6 mm i.d., 5 µm, Analab corporation, Taiwan) with the mobile phases of 70% ddH_2_O (containing 0.1% TFA), 30% ACN (containing 0.1% TFA) at the flow rate of 1 mL/min and temperature of 25 °C. The UV/Vis detector (UV2000D, Analab, Taiwan) was set at 214 nm. The standard curve is constructed using the chemically synthesized α-MSH peptide.

### 3.5. Antimicrobial Activity Assays

The agar diffusion test was used to evaluate antimicrobial activity of cleaved α-MSH. *E. coli* DH5α harboring plasmid pUC19 was cultured in LB medium (containing 50 μg/mL AM) and incubated at 37 °C and 200 rpm until OD_600_ reaching 0.7. The cultured cells were harvested by centrifugation at 7500× *g*, 4 °C, 10 min to obtain the cell pellets, followed by resuspended with sterile PBS buffer to the OD_600_ of 0.7. Then, 150 µL of resuspended cells solution was added to 5 mL of sterile LB broth containing 0.75% agar, followed by pouring it evenly into a petri dish [43]; 10 µL of α-MSH obtained from pepsin cleavage reaction was added to the surface of the agar, and the petri dish was incubated at 37 °C for 16 h. The antimicrobial activity was determined by measuring the diameter of the clear zone around the testing spot.

To estimate the lethal dosage of 50% LD_50_, a ten-fold serial dilution from 100 to 1010 of cleaved α-MSH solution was prepared with Tris buffer (20 mM Tris-HCl pH 2, 250 mM NaCl). The tested *E. coli* DH5α/pUC19 culture was adjusted to 1 × 10^7^ CFU/mL. A volume of 100 μL of the tested bacteria culture was added into 900 μL of different serial diluted cleaved α-MSH solution and mixed gently. Then, 100 μL of the tested sample was evenly spread on an LB agar plate, followed by incubation at 37 °C for 16 h. The CFU was obtained via counting the colony number on the plate in triplicate. The survival rate was defined as
$$\mathrm{Survival}\;\mathrm{rate}(\%)=\frac{\mathrm{CFU}\;\mathrm{of}\;E\;.coli\;\mathrm{treated}\;\mathrm{with}\;\mathrm{various}\;\mathrm{concentrations}\;\mathrm{of}\;\mathrm{\alpha -MSH}}{\mathrm{CFU}\;\mathrm{of}\;E\;.coli\;\mathrm{treated}\;\mathrm{with}\;\mathrm{blank}\;\mathrm{buffer}}$$

LD_50_ was defined as the dose of α-MSH where the amount of cells remains half of the CFU of the original cells [44].

### 3.6. α-MSH Activity Assay

The mouse melanoma B16-F10 cell line (BCRC 60031, Bioresource Collection and Research Center (BCRC), Hsinchu City, Taiwan) was cultured in Dulbecco’s modified minimal essential medium (DMEM, Gibco, Waltham, MA, USA) with 10% fetal bovine serum (FBS) in a humidified atmosphere with 5% CO_2_ at 37 °C, and the medium was changed every two days for the following assays. To assess the cell proliferation and viability [45], B16-F10 cells were plated in 6-well plates at a density of 2 × 10^4^ cells per well (2 mL per well) for attachment first. After a 12 h incubation to allow for cell attachment, the conditioned medium was removed. Then, different concentrations of pepsin cleaved α-MSH samples (0, 10, 50, 100, 200, and 500 ng/mL) were added to the culture medium (2 mL per well), and the plates were incubated for two additional days. Cell proliferation was quantified by performing a Premix WST-1 Cell Proliferation Assay System (Takara, Japan). Cell proliferation was determined by measuring the optical density (OD) after the reaction for 3 h by recording the absorbance at 450 nm using a plate reader (Multiskan Go, Thermo Fisher Scientific, Waltham, MA, USA). Cell viability was determined by the trypan blue exclusion method and counted using a hemocytometer under a microscope. To quantify melanin synthesis induced by α-MSH [45], B16F10 cells were plated in 6-well plates at a density of 2 × 10^4^ cells per well (2 mL per well). After cell attachment to the plates, the cells were treated with different concentrations of α-MSH samples (0, 10, 50, 100, 200, and 500 ng/mL) for 48 h. After treatment, the cells were harvested and washed by Dulbecco’s phosphate-buffered saline (Thermo Fisher Scientific, Waltham, MA, USA) three times. After washing, the cell pellets were resuspended with 0.1 mL M-PERTM mammalian protein extraction reagent (catalog number: 78501, Thermo Fisher Scientific, Waltham, MA, USA) to lyse cell membranes, and then the mixture was centrifuged at 12,000 rpm for 10 min at 4 °C to separate supernatant and the cell debris that contains melanosomes. Melanin was extracted from the cell debris by adding 0.05 mL 1.0 N NaOH, followed by heating at 80 °C in a dry bath for 1 h and shaking every 15 min for 5 s to mix. The relative melanin content was determined by measuring the absorbance at 405 nm. The content of melanin synthesized in cells without α-MSH treatment was set to 100% to compare with the other groups of cells with α-MSH treatment.

### 3.7. Statistical Analysis

All the experiments were performed in triplicate, and data are expressed as means ± standard deviation. The statistical significance was assessed by the multiple comparisons Tukey post hoc analysis of variance (ANOVA) using OriginPro ver.9.0 (OriginLab Corporation, Northampton, MA, USA). The results were considered statistically significant differences at *p*-values < 0.05.

## Figures and Tables

**Figure 1 molecules-26-06207-f001:**
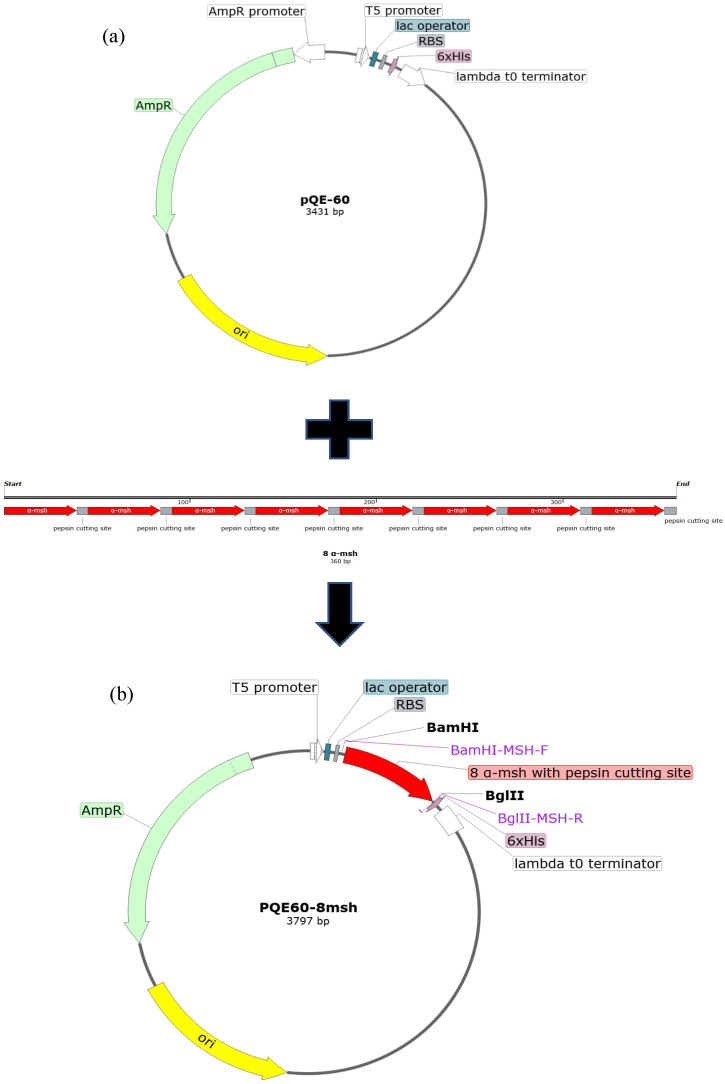
The plasmid map of pQE60-8*msh*. (**a**) Schematic diagram of the gene sequence 8*msh* encoding 8 α-MSH tandem repeat and the expression vector pQE60. Pepsin cutting sites (Gly-Phe, grey boxes) are introduced between the genes of *msh* (red arrows). The total length of the synthetic gene 8*msh* is 260 bp. (**b**) 8*msh* gene flanked with BamHI and BglII restriction sites on 5′- and 3′-ends, respectively, is subcloned into the multiple cloning sites of pQE60 with the fusion of 6xHis-tag at the C-terminus of 8 α-MSH tandem repeat.

**Figure 2 molecules-26-06207-f002:**
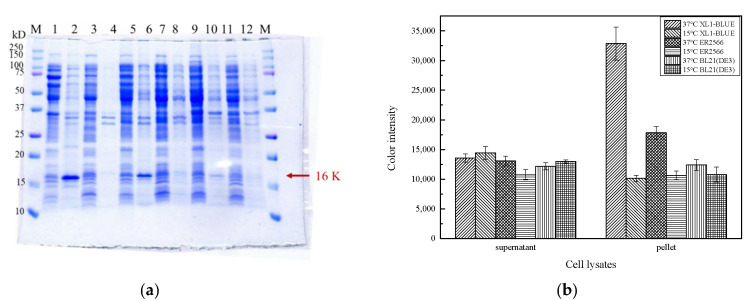
Comparison of different hosts for overexpression of 8*msh*. (**a**) SGS-PAGE gel showing the yield of 8 α-MSH fusion protein, where Lane M—marker (kDa); Lane 1–2—supernatant of cell lysate, pellet of cell lysate/37 °C induction (XL1-Blue); Lane 3–4—supernatant of cell lysate, pellet of cell lysate/15 °C (XL1-Blue); Lane 5–6—supernatant of cell lysate, pellet of cell lysate/37 °C (ER2566); Lane 7–8—supernatant of cell lysate, pellet of cell lysate/15 °C (ER2566); Lane 9–10—supernatant of cell lysate, pellet of cell lysate/37 °C (BL21); Lane 11–12—supernatant of cell lysate, pellet of cell lysate/15 °C (BL21). (**b**) Quantitative analysis of the expression level of 8*msh* using ImageJ.

**Figure 3 molecules-26-06207-f003:**
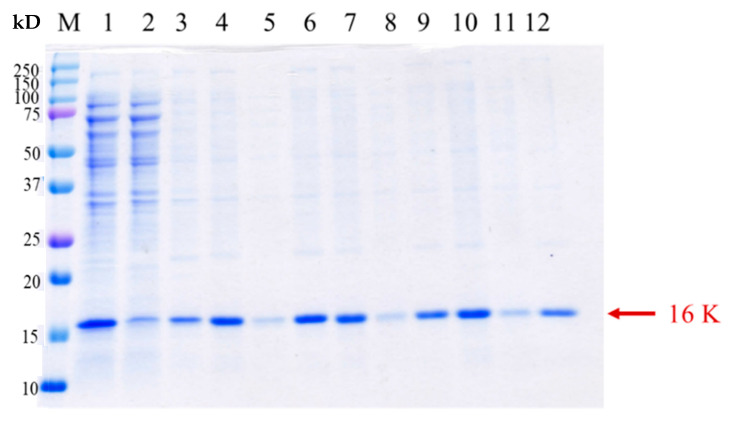
Dissolving 8 α-MSH inclusion body using urea solution (2–6 M). Lane M—marker (kDa); Lane 1–3—original cell lysate, supernatant of cell lysate, pellet of cell lysate; Lane 4–6—2 M urea cell lysate mixture, supernatant of cell lysate, pellet of cell lysate; Lane 7–9—4 M urea cell lysate mixture, supernatant of cell lysate, pellet of cell lysate; Lane 10–12—6 M urea cell lysate mixture, supernatant of cell lysate, pellet of cell lysate.

**Figure 4 molecules-26-06207-f004:**
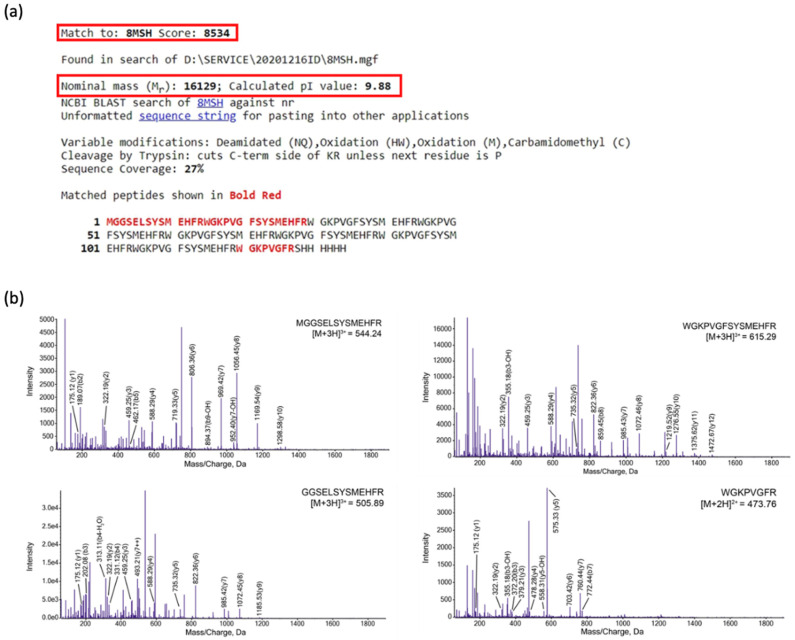
LC-MS/MS analysis demonstrating the identity of 8 α-MSH protein. (**a**) Protein identity fingerprinting of 8 α-MSH using MASCOT. The low sequence coverage (27%) is due to the eight repeats of α-MSH peptide. (**b**) LC-MS/MS spectra demonstrating the identity of the four constituent peptides of trypsin digested 8 α-MSH protein.

**Figure 5 molecules-26-06207-f005:**
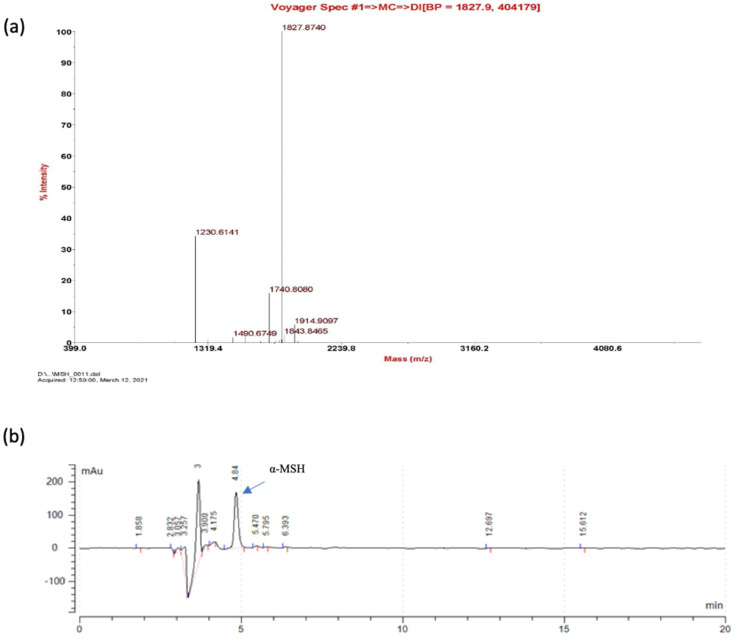
Identification of the pepsin-cleaved α-MSH. (**a**) MALDI-TOF analysis demonstrates the identity of α-MSH with the expected molecular weight (1827.87); (**b**) HPLC analysis of pepsin-cleaved α-MSH peptide. The arrow indicates the peak corresponding to the chemically synthesized α-MSH standard.

**Figure 6 molecules-26-06207-f006:**
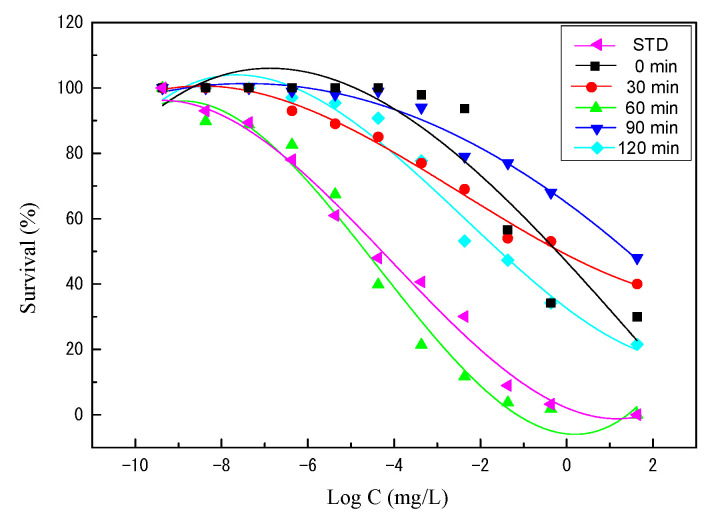
*E. coli* survival rate and LD_50_ determination of pepsin digested 8 α-MSH with different cleavage times (0–120 min). Chemically synthesized α-MSH (STD) is included for comparison.

**Figure 7 molecules-26-06207-f007:**
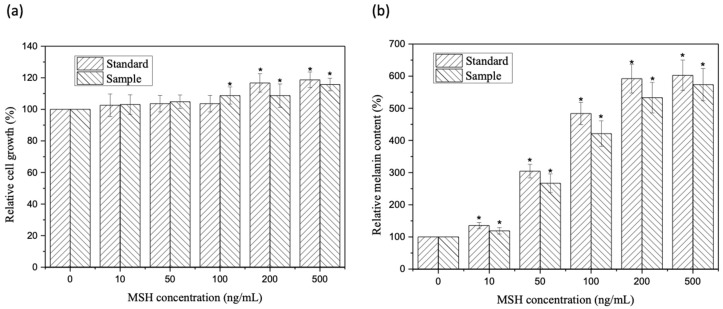
Cytotoxicity assay and melanin stimulation induced by pepsin-cleaved α-MSH. Cytotoxicity assay (**a**) and melanin stimulation assay (**b**) of pepsin-cleaved α-MSH (sample) in comparison with synthesized α-MSH (standard). “*” indicates the results having significant differences with reference (buffer only, MSH = 0) at *p*-values < 0.05.

**Table 1 molecules-26-06207-t001:** The primers used in this study.

Name	Sequence (5′-3′)	Restriction Enzyme
BamHI-msh-F ^a^	CGC GGA TCC TGG CGC GCC GAG CTC TCT TAT AGC	BamHI
BglII-msh-R	GGA AGA TCT AAA ACC CAC TG	BglII

^a^ Source: Genomics BioSci & Tech. Co., Ltd (Taipei, Taiwan). Underline: restriction enzyme site.

**Table 2 molecules-26-06207-t002:** Verification of the target gene 8*msh* using PCR amplification and restriction site cutting.

Host	PCR	Restriction Site
DH5	− ^a^	−
JM109 (DE3)	+	−
BW25113	+	−
EPI300	+	−
W3110	+	−
HB101	+	−
ER2566	+	+
XL1-Blue	+	+
BL21 (DE3)	+	+

^a^: “+” denotes explicit target DNA band with correct size; “−“ denotes unable to detect explicit target DNA band with correct size.

**Table 3 molecules-26-06207-t003:** Effect of different intervals for pepsin cleavage of 8 α-MSH.

Reaction Time (min)	Group I Buffer ^a^	Group II Pepsin ^b^	Group III 8α-MSH with Pepsin ^c^
0	− ^d^	−	+
30	−	−	+
60	−	−	++
90	−	−	+
120	−	−	−

^a^ Group I: 20 mM Tris-HCl, pH 2, 250 mM NaCl. ^b^ Group II: 20 mM Tris-HCl, pH 2, 250 mM NaCl, 0.4% pepsin. ^c^ Group III: 8 α-MSH plus 20 mM Tris-HCl, pH 2, 250 mM NaCl, 0.4% pepsin. ^d^ “−” inhibition zone <0.5 cm; “+” inhibition zone ≤1 cm and >0.5 cm; “++” inhibition zone >1 cm.

**Table 4 molecules-26-06207-t004:** Effect of pepsin cleavage of 8 α-MSH at different pH.

Reaction pH	Group I Buffer ^a^	Group II Pepsin ^b^	Group III 8α-MSH with Pepsin ^c^
2	− ^d^	−	++
4	−	−	+
7	−	−	−

^a^ Group I: 20 mM Tris-HCl, pH 2, 250 mM NaCl. ^b^ Group II: 20 mM Tris-HCl, pH 2, 250 mM NaCl, 0.4% pepsin. ^c^ Group III: 8 α-MSH plus 20 mM Tris-HCl, pH 2, 250 mM NaCl, 0.4% pepsin. ^d^ “−” inhibition zone <0.5 cm; “+” inhibition zone ≤1 cm and > 0.5 cm; “++” inhibition zone >1 cm.

**Table 5 molecules-26-06207-t005:** Lethality response for products from various cleavage intervals (unit: mg/L).

Cleavage Time	LD_0_	LD_20_	LD_50_	LD_100_
Standard	^a^ 6.35 × 10^−5^	2.00 × 10^−3^	1.28 × 10^−2^	1.22 × 10^1^
0	8.93 × 10^−4^	1.54 × 10^−1^	2.46 × 10^0^	6.82 × 10^4^
30	1.23 × 10^−4^	3.60 × 10^−2^	7.66 × 10^−1^	6.07 × 10^4^
60	3.85 × 10^−5^	1.15 × 10^−3^	7.19 × 10^−3^	6.16 × 10^0^
90	4.89 × 10^−4^	3.59 × 10^−1^	1.25 × 10^1^	6.21 × 10^6^
120	2.62 × 10^−4^	2.66 × 10^−2^	3.20 × 10^−1^	3.10 × 10^3^

^a^ Lethal dose parameters are estimated according to the fitting curves in Figure 6.

**Table 6 molecules-26-06207-t006:** Comparison of antibacterial peptide production.

Antimicrobial Peptide	Molecular Weight	Cultivation Conditions	Post-Treatments	Weight Yield (mg/L)	Molar Yield (μmol/L)	Ref.
Cecropin B2	4 kDa	OD_600_ = 0.8 0.1 mM IPTG 25 °C, 16 h	1. Intein cleavage 2. Centrifugal filtration 3. Column separation	58.7	14.68	[33,34]
Cecropin A (C–L)	2.9 kDa	OD_600_ = 0.6~0.8 1.5 mM IPTG 37 °C, 5 h	1. Ni-IMAC 2. SUMO cleavage	17.54	6.05	[35]
LFT33 (LfcinB-thanatin)	4.2 kDa	OD_600_ = 0.6 0.2 mM IPTG 30 °C, 4 h	1. Ni-IMAC 2. HPLC	0.5	0.12	[36]
Bin1b	5.2 kDa	OD_600_ = 0.5 1 mM IPTG 25 °C, 16 h	Ni-IMAC	4.4	0.85	[37]
Mdmcec	4 kDa	OD_600_ = 0.8 0.4 mM IPTG 25 °C, 9 h	1. Ni-IMAC 2. Centrifugal filtration 3. HPLC	11.2	2.80	[38]
α-MSH	1.8 kDa	OD_600_ = 0.4 0.9 mM IPTG 37 °C, 16 h	Pepsin cleavage	42.9	23.83	This study

**Table 7 molecules-26-06207-t007:** The strains and plasmid used in this study.

Strain or Plasmid	Genotype and Relevant Characteristics	Source
BL21 (DE3)	F^−^ ompT gal dcm lon hsdS_B_ (r_B_^−^m_B_^−^) λ^−^ (DE3 [lacI lacUV5-T7 gene ind1 sam7 nin5])	Novagen, WI, USA
BW25113	F^−^ DE(araD-araB)567 lacZ4787(del)::rrnB-3 LAM^−^ rph-1 DE(rhaD-rhaB)568 hsdR514	Coli Genetics Stock Center, New Haven, CT, USA
DH5α	F^−^ endA1 glnV44 thi-1 recA1 relA1 gyrA96 deoR nupG Φ80dlacZΔM15 Δ(lacZYA-argF)U169 hsdR17(rK^−^mK^+^) λ^−^	Novagen, WI, USA
ER2566	F^−^ λ^−^ fhuA2 [lon] ompT lacZ::T7 gene1 gal sulA11 Δ(mcrC-mrr)114::IS10R(mcr-73::miniTn10-TetS)2 R(zgb-210::Tn10)(TetS) endA1[dcm]	New England Biolabs, Ipswich, MA, USA
EPI300	F´[proAB^+^ lacI^q^ lacZΔM15 zzf::Tn10(Tet^R^)] fhuA2 glnV Δ(lac-proAB) thi-1 Δ(hsdS-mcrB)5	Lucigen, WI, USA
HB101	F^−^, thi-1, hsdS20 (r_B_^–^, m_B_^–^), supE44, recA13, ara-14, leuB6, proA2, lacY1, galK2, rpsL20 (str^r^), xyl-5, mtl-1	Promega, WI, USA
JM109 (DE3)	endA1recA1 gyrA96 hsdR17 (r_k_^−^, r_k_^+^) relA1 supE44 λ^−^ Δ(lac-proAB) [F´ traD36 proAB lacI^q^ ZΔM15], lDE3	Promega, WI, USA
W3110	F^−^ λ^−^ rph-1 INV(rrnD, rrnE)	New England Biolabs, Ipswich, MA, USA
XL1-Blue	recA1 endA1 gyrA96 thi-1 hsdR17 supE44 relA1 lac [F´ proAB lacI^q^ Z∆M15 Tn10 (Tet^r^)]	Agilent, Santa Clara, CA, USA
pQE60	bacterial vectors for inducible expression of N-terminally T5-tagged protein	Addgene, Teddington, UK
pUC19	bacterial vectors with lac promoter and ampicillin resistant marker	Addgene, Teddington, UK

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
