# Peer review of "Construction of a Tandem Repeat Peptide Sequence with Pepsin Cutting Sites to Produce Recombinant α-Melanocyte-Stimulating Hormone"

_molecules, 2021, doi:10.3390/molecules26206207_

Round 1
Reviewer 1 Report
The manuscript by Dai-Lin Jiang et al., describes biological preparation of an a-melanocyte stimulating hormone (a-MSH). Authors designed recombinant protein composed of 8 tandem repeats of a-MSH for E. coli expression. Unfortunately, their trials failed to get recombinant protein as a soluble protein., however, they prepared soluble a-MSH by forcefully pepsin digestion of insoluble recombinant protein. Then, they investigated biological activities of prepared a-MSH without purification, and they demonstrated that crude a-MSH had similar biological activities to chemically synthesized a-MSH.
I have to point out the following several problems of this manuscript.
- Because a-MSH is a short peptide, chemical synthesis method seems to be easier than recombinant expression method. Authors should refer to the reports about chemical synthesis of a-MSH or its derivatives with proper references, then claim the advantages of recombinant expression method.
- I could not find pepsin-cleavage site in the sequence of recombinant protein. Is the endogenous glutamic acid included in pepsin-cleavage site? Authors should explain the molecular design of recombinant protein more detail.
- I could not understand why 8msh is nontoxic to the host E. coli cells. Authors should explain the mechanism of antimicrobial activity of a-MSH and 8msh in detail.
- There is no detail description about how to determine the concentration of a-MSH after pepsin digestion. Related to that, the purity of a-MSH after pepsin digestion should be shown in number.
- Although authors used a-MSH after pepsin digestion without purification, I think the purification is necessary for detail biological assays. Related to that, “Pepsin cleavage” is not purification in Table 6 (just preparation of monomeric a-MSH). Authors should not mix up.
Due to the above reasons, I do not recommend this manuscript in Molecules.
Author Response
1. Because a-MSH is a short peptide, chemical synthesis method seems to be easier than recombinant expression method. Authors should refer to the reports about chemical synthesis of a-MSH or its derivatives with proper references, then claim the advantages of recombinant expression method.
A: To the best of our knowledge, the manuscript presents the first successful procedure of producing α-MSH using recombinant protein techniques. Traditional approaches using chemical synthesis to produce α-MSH peptides were utilized for research purposes, such as the effect of specific amino acids by mutagenesis, of which minimum amounts of peptides are required. Because the cost of chemical synthesis is extremely high, 1mg of α-MSH with the price of nearly 200 USD, it is impractical to apply the chemically synthesized peptides in cosmetics, animal feeds and antimicrobial agents. As a consequence, this is the reason why we aim to develop an economical procedure using protein engineering and fermentation to fulfill the industrial needs. We added a sentence to highlight the significance of recombinant protein for industrial purpose (Line 60). Two references of α-MSH production using chemical synthesis are included as the reviewer suggested.
(1) Synthesis and structure-function studies of melanocyte stimulating hormone analogues modified in the 2 and 4(7) positions: comparison of activities on frog skin melanophores and melanoma adenylate cyclase.
- J. Hruby, T. K. Sawyer, Y. C. Yang, M. D. Bregman, M. E. Hadley, C. B. Heward.
- Med. Chem., 1980, 23, 1432-1437
(2) Synthesis of α-and β-melanocyte stimulating hormones
- C.S. Yang, V. J. Hruby, C. B. Heward, M. E. Hadley
Int. J. Peptide Protein Res.,1980, 15, 130-138
2. I could not find pepsin-cleavage site in the sequence of recombinant protein. Is the endogenous glutamic acid included in pepsin-cleavage site? Authors should explain the molecular design of recombinant protein more detail.
A: In fact, pepsin is a digestive protease in gastrointestinal tract discovered by Theodor Schwann. The enzyme is produced by gastric chief cells and can cleave proteins into small peptides at pH = 1.5~3.5. The recognition sites for pepsin cleavage include Phe-Val, Gln-His, Glu-Ala, Ala-Leu, Leu0Tyr, Tyr-Leu, Gly-Phe, Phe-Phe and Phe-Tyr. In our research study, the reason why we chose Gly-Phe (GF) cleavage site between two α-MSH repeats is because the amino acid sequence motif (GF) is not present in α-MSH and pepsin digestion could only produce α-MSH fragments but would not cleave the peptide itself. The sequence of pepsin cleavage site has been supplemented in Line 79 and in Fig.1.
(1) Florkin, M. Discovery of pepsin by Theodor Schwann. Rev Med Liege, 1957, 12, 139-44.
(2) Burrell, M.M. Enzymes of molecular biology. Humana Press, 1993.
3. I could not understand why 8msh is nontoxic to the host E. coli cells. Authors should explain the mechanism of antimicrobial activity of a-MSH and 8msh in detail.
A: As shown by the results, overexpression of 8msh resulted in insoluble inclusion body. Because of the insolubility, it significantly reduced the cytotoxicity to E. coli. A possible explanation has been added in Line 272.
4. There is no detail description about how to determine the concentration of a-MSH after pepsin digestion. Related to that, the purity of a-MSH after pepsin digestion should be shown in number.
A: The concentration of pepsin-cleaved α-MSH was determined by HPLC using synthetic α-MSH as the standard. As the relative activity of cleaved α-MSH is comparably weaker than synthetic α-MSH, we used bioactivity to define the purity of α-MSH. In Fig 7(b), we compared melanin production induced by pepsin-cleaved α-MSH and standard, showing 0.895 ± 0.03 relative bioactivity. It is thus assumed that the purity of pepsin-cleaved α-MSH in our study is approximately 90%. The purity of cleaved α-MSH has been specified in Line 252.
5. Although authors used a-MSH after pepsin digestion without purification, I think the purification is necessary for detail biological assays. Related to that, “Pepsin cleavage” is not purification in Table 6 (just preparation of monomeric a-MSH). Authors should not mix up.
A: We thank the reviewer’s comment. To avoid confusion, we changed “purification” to “post-treatments”.
Reviewer 2 Report
The authors present a detailed and novel expression and purification methodology for the 13 amino acid melanocyte stimulating hormone. Here they introduce αMSH as a tandem repeat of 8 genes with pepsin cleavage sites between each one. Although the authors initial intention appeared to be to use IMAC for purification, the resulting protein was insoluble (but of reasonable purity based on SDS-PAGE). Upon cleavage with pepsin, the soluble αMSH was produced and the activity of the resulting peptide was compared to a chemically synthesise standard.
The method presented is a facile way to produce a challenging target. I do have a few questions for the authors to consider:
1) It is shown that the insoluble fraction is of good purity (~90%), however, as this protein is produced in E.coli, would it not be necessary for further purification steps to make the resulting peptide usable as a potential therapeutic? Did the authors use HPLC to gain high-purity after the pepsin cleavage? It is noted in the melanin production assay that the E. coli produced sample was not as good as the chemically treated one.
2) Pepsin cleavage was chosen as it is an enzyme present in the digestive system; however, it seems that too little cleavage or too long of cleavage has a significant effect on the activity of the peptide - how would this be controllable if the fusion protein was administered orally?
3) The lack of solubility in 6M Urea suggests to me that a different chaotropic agent is needed - why did the authors not try GdnHCl (for purification of inclusion bodies, GdnHCl is often employed).
Author Response
1. It is shown that the insoluble fraction is of good purity (~90%), however, as this protein is produced in E. coli, would it not be necessary for further purification steps to make the resulting peptide usable as a potential therapeutic? Did the authors use HPLC to gain high-purity after the pepsin cleavage? It is noted in the melanin production assay that the E. coli produced sample was not as good as the chemically treated one.
A: In fact, this study aimed to develop a procedure for mass production of α-MSH peptide. We quantified the yield of α-MSH using HPLC as mentioned in Materials and Methods (Line 324) The relative purity is approximately 90% calculated by the bioactivity of melanin production in comparison with synthetic standard (Fig. 7b). At this stage, we think this purity is sufficient as an additive for animal feeds or cosmetics applications. For therapeutic application, it probably needs further polish to attain higher level of safety. The significance of recombinant techniques has been emphasized in Line 281.
2. Pepsin cleavage was chosen as it is an enzyme present in the digestive system; however, it seems that too little cleavage or too long of cleavage has a significant effect on the activity of the peptide - how would this be controllable if the fusion protein was administered orally?
A: Oral administration for pepsin digestion in gastrointestinal tract proposes a potential procedure for peptide production. However, for therapeutic application, it requires further purification and clinical trials to ensure the safety. To avoid any misunderstanding, we deleted the sentence accordingly.
3. The lack of solubility in 6M Urea suggests to me that a different chaotropic agent is needed - why did the authors not try GdnHCl (for purification of inclusion bodies, GdnHCl is often employed).
A: We thank the reviewer’s suggestion. We discovered that 8M urea failed to dissolve 8α-MSH. Even if higher concentration of urea can dissolve the inclusion body, it requires dialysis to reduce the concentration of urea. As the purity of insoluble fraction of cell lysate was considerably good, we proceeded the following procedures to obtain α-MSH peptide and did not try GdnHCl. In the future, if soluble 8α-MSH is required, we will definitely try the alternative protein denaturant.
Round 2
Reviewer 1 Report
The manuscript is partly revised according to the comments. However, I have still question to be revised in this manuscript as follows:
- I cannot understand why bacterial expression system has superior cost-performance to the chemical synthesis method. The cost in chemical synthesis of peptide highly depends on the synthesis scale. The large synthesis scale can decrease the cost.
- I could not follow the experimental design of 8msh protein expression. Why do authors think that 8msh will not decrease cytotoxicity to host cells? Did authors aim that expressed protein forms inclusion body in the step of 8msh protein design? Although authors did not use His-tag for purification, is His-tag necessary for 8msh protein design?
- In previous comment about the purity of pepsin-digested a-MSH, I want to confirm the purity based on HPLC analysis, not based on biological activity, because the biological activity just provides indirect information about purity. At least, His-tag fragment has to be included as impurity.
- In previous comment, I want to strongly suggest purifying pepsin-digested a-MSH by chromatography (HPLC, ion-exchange chromatography, and so on), because there is possibility of side effects derived from impurities such as His-tag fragment and partially fragmented a-MSH.
Due to the above reasons, this research requires some additional experiments.